# Entanglement Dynamics and Classical Complexity

**DOI:** 10.3390/e25010097

**Published:** 2023-01-03

**Authors:** Jiaozi Wang, Barbara Dietz, Dario Rosa, Giuliano Benenti

**Affiliations:** 1Department of Physics, University of Osnabrück, D-49069 Osnabrück, Germany; 2Center for Theoretical Physics of Complex Systems, Institute for Basic Science (IBS), Daejeon 34126, Republic of Korea; 3Basic Science Program, Korea University of Science and Technology (UST), Daejeon 34113, Republic of Korea; 4Center for Nonlinear and Complex Systems, Dipartimento di Scienza e Alta Tecnologia, Università degli Studi dell’Insubria, Via Valleggio 11, 22100 Como, Italy; 5Istituto Nazionale di Fisica Nucleare, Sezione di Milano, Via Celoria 16, 20133 Milano, Italy; 6NEST, Istituto Nanoscienze-CNR, 56126 Pisa, Italy

**Keywords:** quantum complexity, quantum to classical transition

## Abstract

We study the dynamical generation of entanglement for a two-body interacting system, starting from a separable coherent state. We show analytically that in the quasiclassical regime the entanglement growth rate can be simply computed by means of the underlying classical dynamics. Furthermore, this rate is given by the Kolmogorov–Sinai entropy, which characterizes the dynamical complexity of classical motion. Our results, illustrated by numerical simulations on a model of coupled rotators, establish in the quasiclassical regime a link between the generation of entanglement, a purely quantum phenomenon, and classical complexity.

## 1. Introduction

The characterization of complexity in quantum systems is a key problem, not only for fundamental reasons but also for the development of quantum technologies [1,2,3]. While for classical dynamical systems a well-established notion of complexity exists, based on Kolmogorov–Sinai (KS) entropy [4], which in turn is related to the exponential instability of orbits, in the quantum realm the measure of complexity has proven to be an elusive problem.

First, we cannot *sic et simpliciter* use trajectories, due the Heisenberg uncertainty principle. To circumvent such problem, phase-space approaches have been proposed [5,6,7,8,9,10,11,12,13,14,15,16,17,18,19,20], based on the evolution of phase space distributions. Second, entanglement, the key resource in the quest for quantum advantage, is peculiar to quantum composite systems and therefore is a source of quantum complexity without a classical analogue. Since for pure bipartite systems the reduced von Neumann entropy, known as entanglement entropy, is the well-established measure of entanglement [21], it is interesting to investigate whether its growth in a dynamical system is related to the KS entropy of the underlying classical dynamics [22].

For bosonic systems with an unstable quadratic Hamiltonian, entanglement entropy grows linearly in time, with a rate upper bounded by the KS entropy, the bound being saturated under suitable conditions on the size of the bipartitions [23]. The question then arises, whether the entanglement growth of chaotic quantum systems in the quasiclassical regime is also determined by the KS entropy. This issue was investigated more than two decades ago, with numerical results suggesting that the entanglement generation rate is given by the KS entropy [24]. On the other hand, such results were obtained in the weakly chaotic regime, with coexistence of chaotic seas and tori, while another study in the strongly chaotic region, where the effect of tori is negligible, showed instead no increase of the entanglement production rate upon an increase of the maximum Lyapunov exponents [25]. This apparent contradiction was explained by a quasiclassical calculation for the linear entropy, approximating the entanglement entropy, under the condition of weak coupling between the subsystems in the underlying classical dynamics [26,27]. This work showed that the entanglement growth rate is determined by the minimal value of the three rates given by the standard one deduced from the interaction term and the largest Lyapunov exponents of the two subsystems, respectively.

In this paper, we remove the above restriction on the coupling strength and compare the quantum evolution starting from separable coherent states with the classical evolution of initially Gaussian distributions, of a size determined by the effective Planck constant of the corresponding quantum dynamics. We show that in the quasiclassical regime quantum and classical linear entropy are in agreement and grow with the rate given by the KS entropy of classical dynamics. Our analytical results are illustrated by numerical simulations for a model of kicked-coupled rotators.

This work is dedicated to our friend and colleague Giulio Casati, who has always had a deep interest in understanding the complexity of quantum motion.

## 2. Analytical Results

In this section, we connect, for an overall pure bipartite system, the growth rate of linear entropy to the KS entropy of the classical underlying dynamics. We consider a two body system, whose Hamiltonian reads
(1)$$\hat{H}={\hat{H}}_{1}\left({\hat{q}}_{1},{\hat{p}}_{1}\right)+{\hat{H}}_{2}\left({\hat{q}}_{2},{\hat{p}}_{2}\right)+{\hat{H}}_{12}\left({\hat{q}}_{1},{\hat{p}}_{1},{\hat{q}}_{2},{\hat{p}}_{2}\right).$$The corresponding classical Hamiltonian is written as
(2)$$H\left(q_{1},p_{1},q_{2},p_{2}\right)=H_{1}\left(q_{1},p_{1}\right)+H_{2}\left(q_{2},p_{2}\right)+H_{12}\left(q_{1},p_{1},q_{2},p_{2}\right).$$We compute as entanglement measure the linear entropy (also known as *second Rényi entropy*) of a subsystem (for example, system 1), which is defined as
(3)$$S\left({\hat{\rho }}_{1}\right)=-\ln\left(\mathrm{Tr}\left({\hat{\rho }}_{1}^{2}\right)\right).$$Here ${\hat{\rho }}_{1}$ is the reduced density matrix of the system 1, ${\hat{\rho }}_{1}={\mathrm{Tr}}_{2}\left(\hat{\rho }\right)$, where the partial trace is taken over system 2 and $\hat{\rho }$ is the density matrix of the composite system. Note that equivalently we could have considered system 2, since $S\left({\hat{\rho }}_{2}\right)=S\left({\hat{\rho }}_{1}\right)$, with ${\hat{\rho }}_{2}={\mathrm{Tr}}_{1}\left(\hat{\rho }\right)$. The linear entropy is much more convenient for numerical and analytical investigations than the reduced von Neumann entropy $S_{\mathrm{vN}}\left({\hat{\rho }}_{1}\right)=-\mathrm{Tr}\left({\hat{\rho }}_{1}\ln{\hat{\rho }}_{1}\right)$. At the same time, it is a very useful entanglement probe. Indeed, if the linear entropy of a part is larger than the linear entropy of the overall system, bipartite entanglement exists between that part and the rest of the system. Moreover, for maximally mixed states of a subsystem of dimension *N*, the linear entropy and the reduced von Neumann entropy are both maximized and equal to lnN.

In order to obtain the classical analog of the linear entropy, we make use of the Husimi function [28] of the density matrix $\hat{\rho }$, given by
(4)$$W_{H}\left(\gamma \right)=\frac{1}{{\left(2\pi \hbar \right)}^{2}}\langle \gamma |\hat{\rho }|\gamma \rangle ,$$
where $\gamma =(q_{1},p_{1},q_{2},p_{2})$, |γ〉 denotes the coherent state of the composite system centered at γ, and *ℏ* is the effective Planck constant. In the quasiclassical limit ℏ→0, the trace of ${\hat{\rho }}_{1}^{2}$ can be carried out by making use of the Husimi function $W_{H}^{1}$ of ${\hat{\rho }}_{1}$ as
(5)$$\mathrm{Tr}\left({\hat{\rho }}_{1}^{2}\right)=\int d\gamma_{1}{\left[W_{H}^{1}\left(\gamma_{1}\right)\right]}^{2},$$
where $\gamma_{1}=\left(q_{1},p_{1}\right)$, $|\gamma_{1}\rangle$ denotes the coherent state of system 1 centered at $\gamma_{1}$, and
(6)$$W_{H}^{1}\left(\gamma_{1}\right)=\frac{1}{2\pi \hbar }\langle \gamma_{1}|{\hat{\rho }}_{1}|\gamma_{1}\rangle .$$Furthermore, the reduced density matrix ${\hat{\rho }}_{1}$ can also be obtained in terms of the coherent states of the system 2, denoted by $|\gamma_{2}\rangle$, as
(7)$${\hat{\rho }}_{1}={\mathrm{Tr}}_{2}\left(\hat{\rho }\right)=\frac{1}{2\pi \hbar }\int d\gamma_{2}\langle \gamma_{2}|\hat{\rho }|\gamma_{2}\rangle .$$Substituting Equation (Equation 7) into Equation (Equation 6), we have
(8)$$W_{H}^{1}\left(\gamma_{1}\right)=\int d\gamma_{2}W_{H}\left(\gamma \right),$$
yielding with Equation (Equation 5)
(9)$$\mathrm{Tr}\left({\hat{\rho }}_{1}^{2}\right)=\int d\gamma_{1}{\left|\int W_{H}\left(\gamma \right)d\gamma_{2}\right|}^{2}.$$Hence, we obtain
(10)$$S\left({\hat{\rho }}_{1}\right)=-\ln\left[\int d\gamma_{1}{\left(\int d\gamma_{2}W_{H}\left(\gamma \right)\right)}^{2}\right].$$

After replacing the Husimi function $W_{H}\left(\gamma \right)$ with the classical distribution function ρ(γ), the classical analog of linear entropy can be written as
(11)$$S_{cl}\left(\rho_{1}\right)=-\ln\left[\int d\gamma_{1}{\left(\rho_{re}^{1}\left(\gamma_{1}\right)\right)}^{2}\right],$$
where $\rho_{re}^{1}\left(\gamma_{1}\right)$ indicates the marginal distribution function of $\gamma_{1}$,
(12)$$\rho_{re}^{1}\left(\gamma_{1}\right)=\int d\gamma_{2}\rho \left(\gamma \right).$$It is expected that
(13)$$S\left({\hat{\rho }}_{1}\right)\approx S_{cl}\left(\rho_{1}\right)$$
holds in the quasiclassical limit in which the effective Planck constant ℏ→0.

An explicit expression can be derived for the classical entropy $S_{cl}\left(\rho_{1}\right)$ as follows. We consider the initial state as the “most classical” state, that is, a coherent state |γ〉, whose corresponding classical distribution function can be written as
(14)$$\rho_{0}\left(\gamma \right)=\frac{1}{{\left(\pi \hbar_{c}\right)}^{2}}\exp\left(-\frac{1}{\hbar_{c}}{\left|\gamma -\gamma^{0}\right|}^{2}\right),$$
which has a Gaussian form whose center is denoted by $\gamma^{0}$, $\hbar_{c}=\hbar$ is chosen to be the same as the effective Planck constant in the quantum case, and $|\gamma -\gamma^{0}|$ indicates the norm of the vector $\delta \gamma =\gamma -\gamma^{0}$. In the quasiclassical limit, one has $\hbar_{c}\to 0$, which means that, for times smaller than the Ehrenfest time scale $t_{E}$ (with $t_{E}\to \infty$ as $\hbar_{c}\to 0$), almost all the states in the ensemble remain close to the center $\gamma^{0}\left(t\right)$. This implies that the distribution of states at time *t*, $\rho_{t}\left(\gamma \right)$, is significantly different from zero only for small |δγ|. In this case, the time evolution of δγ is determined by the so-called stability matrix
(15)$$M_{t}^{ij}=\frac{\partial (\delta \gamma_{i}\left(t\right))}{\partial (\delta \gamma_{j}\left(0\right))}{|}_{\delta \gamma (0)=0},$$
with
(16)$$\delta \gamma \left(t\right)=M_{t}\delta \gamma \left(0\right).$$As the classical linear entropy is independent of the coordinate’s origin, for the convenience of the following discussion, we choose the position of the center $\gamma^{0}\left(t\right)$ as the origin of coordinates. In this local coordinate system along $\gamma^{0}\left(t\right)$, we can replace $\delta \gamma \left(t\right)=\gamma \left(t\right)-\gamma^{0}\left(t\right)$ by γ(t).

Then making use of Liouville’s theorem, the distribution at time *t* can be written as
(17)$$\rho_{t}\left(\gamma \right)=\rho_{0}\left(M_{t}^{-1}\gamma \right),$$
and therefore
(18)$$\rho_{t}\left(\gamma \right)=\frac{1}{{\left(\pi \hbar_{c}\right)}^{2}}\exp\left(-\frac{1}{\hbar_{c}}{\left|M_{t}^{-1}\gamma \right|}^{2}\right).$$Using the positive definite symmetric matrix
(19)$$A_{t}\equiv {\left(M_{t}^{-1}\right)}^{T}M_{t}^{-1},$$
the density distribution at time *t* can be written as
(20)$$\rho_{t}\left(x\right)=\frac{1}{{\left(\pi \hbar_{c}\right)}^{2}}\exp\left(-\frac{1}{\hbar_{c}}\sum_{i,j=1}^{4}x_{i}A_{t}^{ij}\left(t\right)x_{j}\right),$$
which is a Gaussian distribution, with x corresponding to γ, that is, $\left(x_{1},x_{2},x_{3},x_{4}\right)=\left(q_{1},p_{1},q_{2},p_{2}\right)$. In order to calculate the classical linear entropy, we first calculate the marginal distribution function of $\rho_{t}\left(x\right)$ for system 1:(21)$$\rho_{t}^{1}\left(x_{1},x_{2}\right)=\int \rho_{t}\left(x_{1},x_{2},x_{3},x_{4}\right)dx_{3}dx_{4}.$$Then the classical linear entropy at time *t* can be written as
(22)$$S_{cl}\left(\rho_{t}\right)=-\ln\left[\int dx_{1}dx_{2}{\left(\rho_{t}^{1}\left(x_{1},x_{2}\right)\right)}^{2}\right].$$As outlined in the Appendix A, by writing $A_{t}$ in block form,
(23)$$A_{t}=\left(\begin{array}{cc} \hat{a}\; & \hat{b}\\ {\hat{b}}^{T}\; & \hat{d} \end{array}\right)\;,$$
we obtain our first main result
(24)$$S_{cl}\left(\rho_{t}\right)=\ln\left(2\pi \hbar \right)+\frac{1}{2}\ln\left[\det\left(\hat{d}\right)\right].$$In the derivation we restrict to the case of a two-particle system, however, the generalization to *N* particles is straightforward.

In order to compute $\det\hat{d}$, we sum the eigenvalues of the operator $\hat{d}$ (denoted by $d_{k}$, in order of descending energy), which are in close relation to the eigenvalues of $A_{t}$. We diagonalize the symmetric matrix $A_{t}$ as
(25)$$A_{t}=V\mathrm{diag}\left\{A_{1},A_{2},A_{3},A_{4}\right\}V^{T},$$
where diag indicates a diagonal matrix, $A_{k}$ is the *k*-th eigenvalue of $A_{t}$, and V is an orthogonal matrix. If the system is chaotic, $A_{k}\propto e^{2\lambda_{k}t}$, where $\lambda_{k}$ is the *k*-th Lyapunov exponent, with $\lambda_{1}>\lambda_{2}>0>\lambda_{3}>\lambda_{4}$, and $\lambda_{3}=-\lambda_{2}$, $\lambda_{4}=-\lambda_{1}$.

Hence, in the typical case in which the eigenvectors (denoted by $|A_{k}\rangle$, k=1,2) of $A_{t}$ corresponding to the eigenvalues $A_{1}$ and $A_{2}$ have non-zero components within the Hilbert space of system 2, we have
(26)$$d_{1}\propto e^{2\lambda_{1}t},\;d_{2}\propto e^{2\lambda_{2}t}.$$As a result,
(27)$$\det\hat{d}\propto \exp2(\lambda_{1}+\lambda_{2})t,$$
which directly leads to
(28)$$S_{cl}\left(\rho_{t}\right)-S_{cl}\left(\rho_{0}\right)=\left(\lambda_{1}+\lambda_{2}\right)t,$$
indicating that the growth rate of the linear entropy is given by the Kolmogorov–Sinai entropy of the overall system, which is the second main result of our work.

## 3. Numerical Results

In this section, we numerically illustrate the prediction of equivalence between the classical and quantum growth of linear entropies, Equation (Equation 13), as well as the growth as predicted in Equation (Equation 28), by means of a two-body system which has a clearly defined classical counterpart. More specifically, we consider two coupled rotators (or coupled tops) [29,30], with respective angular momentum operators $\hat{S}={\left({\hat{S}}_{x},{\hat{S}}_{y},{\hat{S}}_{z}\right)}^{T}$ and $\hat{L}={\left({\hat{L}}_{x},{\hat{L}}_{y},{\hat{L}}_{z}\right)}^{T}$, and a time-dependent Hamiltonian with kicked interaction:(29)$$\hat{H}=\frac{a}{j}\left({\hat{S}}_{z}+{\hat{L}}_{z}\right)+\frac{c}{j^{2}}{\hat{S}}_{x}{\hat{L}}_{x}\sum_{n=-\infty }^{\infty }\delta \left(t-n\right),$$
where *j* is the (half-integer or integer) total angular momentum quantum number of both tops. The Hamiltonian possesses constants of motion, ${\hat{S}}^{2}$ and ${\hat{L}}^{2}$. The Hilbert space is expanded by making use of $|s,m_{s},l,m_{l}\rangle \equiv |s,m_{s}\rangle ⊗|l,m_{l}\rangle$, which are the joint eigenvectors of ${\hat{S}}^{2},{\hat{S}}_{z},{\hat{L}}^{2},{\hat{L}}_{z}$,
(30)$$\begin{array}{c} {\hat{S}}^{2}|s,m_{s},l,m_{l}\rangle =s\left(s+1\right)|s,m_{s},l,m_{l}\rangle ,\\ {\hat{S}}_{z}|s,m_{s},l,m_{l}\rangle =m_{s}|s,m_{s},l,m_{l}\rangle ,\\ {\hat{L}}^{2}|s,m_{s},l,m_{l}\rangle =l\left(l+1\right)|s,m_{s},l,m_{l}\rangle ,\\ {\hat{L}}_{z}|s,m_{s},l,m_{l}\rangle =l_{s}|s,m_{s},l,m_{l}\rangle . \end{array}$$
where $m_{s}\in \left\{-s,-s+1,\cdots ,s-1,s\right\}$, and $l_{s}\in \left\{-l,-l+1,\cdots ,l-1,l\right\}$. Here we choose s=l=j.

The Floquet operator, that is the unitary evolution operator between consecutive kicks, can be written as
(31)$$\hat{F}=\exp\left[-ia\left({\hat{S}}_{z}+{\hat{L}}_{z}\right)\right]\exp\left[-i\frac{c}{j}{\hat{S}}_{x}{\hat{L}}_{x}\right].$$The classical counterpart can be obtained by taking the quasiclassical limit $\hbar =\frac{1}{j}\to 0$. Introducing the rescaled angular momenta ${\hat{S}}_{k}=\frac{{\hat{S}}_{k}}{j}$ and ${\hat{L}}_{k}=\frac{{\hat{L}}_{k}}{j}$, and considering the quasiclassical limit j→∞, yields the classical analog of the model,
(32)$$H_{c}=a\left(S_{z}+L_{z}\right)+cS_{x}L_{x}\sum_{n=-\infty }^{\infty }\delta \left(t-n\right),$$
where $S_{x}^{2}+S_{y}^{2}+S_{z}^{2}=L_{x}^{2}+L_{y}^{2}+L_{z}^{2}=1$. Depending on the coupling strength the classical motion can be either chaotic or nearly-integrable, as shown by the three-dimensional Poincaré surfaces of sections of Figure 1.

In the numerical simulations of both the quantum and classical cases, the linear entropy is averaged over $N_{p}$ different initial states. In the quantum case, we consider the initial states $|\theta_{1},\phi_{1},\theta_{2},\phi_{2}\rangle \equiv |\theta_{1},\phi_{1}\rangle ⊗|\theta_{2},\phi_{2}\rangle$, where $|\theta_{1},\phi_{1}\rangle$ and $|\theta_{2},\phi_{2}\rangle$ indicate the spin coherent state of the first rotator,
(33)$$|\theta_{1},\phi_{1}\rangle =e^{i\theta_{1}{\hat{S}}_{z}}e^{i\phi_{1}{\hat{S}}_{y}}|j,j\rangle ,$$
and an analogous expression holds for the second rotor. Then, the quantum averaged linear entropy is calculated as follows,
(34)$${\bar{S}}_{q}\left(t\right)=\frac{1}{N_{p}}\sum_{p}\mathrm{Tr}\left({\left({\hat{\rho }}_{1}^{p}\left(t\right)\right)}^{2}\right),$$
where
(35)$${\hat{\rho }}_{1}^{p}\left(t\right)={\mathrm{Tr}}_{2}({\hat{F}}^{t}|\theta_{1}^{p},\phi_{1}^{p},\theta_{2}^{p},\phi_{2}^{p}\rangle \langle \theta_{1}^{p},\phi_{1}^{p},\theta_{2}^{p},\phi_{2}^{p}|{\left({\hat{F}}^{†}\right)}^{t}),$$
and $\left(\theta_{1}^{p},\phi_{1}^{p},\theta_{2}^{p},\phi_{2}^{p}\right)$ are chosen randomly. In the classical case, we consider an initial ensemble of Gaussian states,
(36)$$\begin{array}{c} \rho_{0}\left(\theta_{1}^{\prime },\phi_{1}^{\prime },\theta_{2}^{\prime },\phi_{2}^{\prime }\right)=A\exp\left(-\frac{{\left(\theta_{1}^{\prime }-\theta_{1}\right)}^{2}}{\hbar_{c}}-\frac{\sin^{2}\left(\theta_{1}\right){\left(\phi_{1}^{\prime }-\phi_{1}\right)}^{2}}{\hbar_{c}}\right)\\ \times \exp\left(-\frac{{\left(\theta_{2}^{\prime }-\theta_{2}\right)}^{2}}{\hbar_{c}}-\frac{\sin^{2}\left(\theta_{2}\right){\left(\phi_{2}^{\prime }-\phi_{2}\right)}^{2}}{\hbar_{c}}\right), \end{array}$$
which in case of $\hbar_{c}\to 0$ can be written in terms of canonical variables $\left(q_{1},p_{1},q_{2},p_{2}\right)=\left(\phi_{1},\cos\theta_{1},\phi_{2},\cos\theta_{2}\right)$ as
(37)$$\begin{array}{c} \rho_{0}\left(q_{1}^{\prime },p_{1}^{\prime },q_{2}^{\prime },p_{2}^{\prime }\right)=A^{\prime }\exp\left(-\frac{{\left(q_{1}^{\prime }-q_{1}\right)}^{2}}{\hbar_{c}\left(1-p_{1}^{2}\right)}-\frac{\left(1-p_{1}^{2}\right){\left(p_{1}^{\prime }-p_{1}\right)}^{2}}{\hbar_{c}}\right)\\ \times \exp\left(-\frac{{\left(q_{2}^{\prime }-q_{2}\right)}^{2}}{\hbar_{c}\left(1-p_{2}^{2}\right)}-\frac{\left(1-p_{2}^{2}\right){\left(p_{2}^{\prime }-p_{2}\right)}^{2}}{\hbar_{c}}\right). \end{array}$$Here *A* and $A^{\prime }$ are normalization constants. Then the classical averaged linear entropy is calculated as
(38)$${\bar{S}}_{cl}\left(t\right)=\frac{1}{N_{p}}\sum_{p}S_{cl}\left(\rho_{t}^{p}\right),$$
where $S_{cl}\left(\rho_{t}^{p}\right)$ indicates the classical linear entropy (defined in Equation (Equation 11)), starting from the initial ensemble, centered at $\left(q_{1}^{k},p_{1}^{k},q_{2}^{k},p_{2}^{k}\right)$. In our numerical simulations, we considered $10^{7}$ trajectories for each initial ensemble, and the integral in Equation (Equation 11) is calculated by summing over the whole phase space with respect to system 1, which is divided into $4\times 10^{6}$ phase cells.

Results for the chaotic regime are shown in Figure 2. Note that $\lambda_{2}$ is comparable to $\lambda_{1}$, and the behavior $S\left(t\right)-S\left(0\right)=\left(\lambda_{1}+\lambda_{2}\right)t$ predicted in Equation (Equation 28) on the basis of a purely classical calculation, can be clearly seen both for quantum and classical simulations. The growth rate, in very good agreement with the KS entropy $\lambda_{1}+\lambda_{2}$, is clearly distinguished from the growth rate given by the largest Lyapunov exponent $\lambda_{1}$ alone. Note that by increasing the coupling strength *c* the entanglement growth rate increases, in accordance with the increase of the classical KS entropy. Moreover, it can be clearly seen that the agreement between the classical and quantum linear entropy extends to longer times as $\hbar =\hbar_{c}$ is reduced.

In Figure 3, we show data in the regular regime with weaker coupling strength, for which invariant tori of the integrable model at c=0 are deformed but survive. The volume occupied by the tori is the largest portion of the phase space and this affects the growth of the linear entropy, which is logarithmic rather than linear. Our numerical results show that, for large enough *ℏ*, the entropy $\bar{S}\left(t\right)\propto \log t^{\alpha }$, with α≈1, while α slowly increases with reducing *ℏ*. Note that the separation between nearby trajectories increases linearly in time for integrable dynamics. Therefore, the number of cells of area *ℏ* occupied in the two-dimensional phase-space for system 1 is proportional to $t^{2}$, leading to the expected growth $\bar{S}\left(t\right)\propto \log t^{2}$. We therefore conjecture that such growth would be achieved in the limit ℏ→0.

## 4. Conclusions

We have shown that in the quasiclassical regime the entanglement growth rate is given by the Kolmogorov–Sinai entropy of the underlying classical dynamics. Note that we are considering initial separable coherent states, so that the quantum wave packet closely follows the underlying classical phase space distribution up to the Ehrenfest time, which diverges as the effective Planck constant ℏ→0. In spite of the lack of entanglement in classical mechanics, our results prove, in the quasiclassical regime, the close connection between entanglement generation and complexity of classical motion. Moreover, our derivation based on purely classical grounds provides an intuitive picture that could hardly be obtained on the basis of purely quantum calculations. Finally, the entanglement growth is linear in the classically chaotic and logarithmic in the regular regime, thus showing the entangling power of chaos.

## Figures and Tables

**Figure 1 entropy-25-00097-f001:**
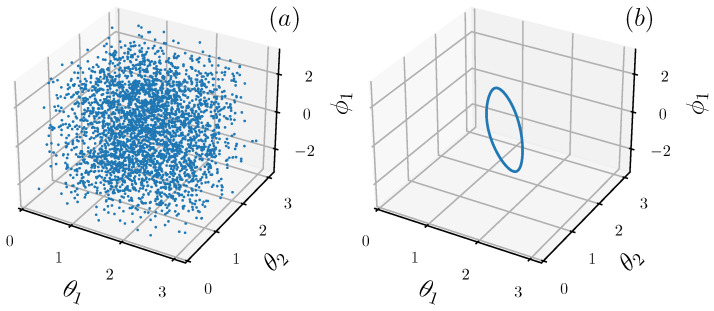
Three-dimensional Poincaré surface of section for the chaotic case (**a**): a=5,c=3 and the near-integrable case (**b**): a=5,c=0.5, where we fix $\phi_{2}=0$. Here we only consider a single trajectory starting from $\left(\theta_{1},\phi_{1},\theta_{2},\phi_{2}\right)=\left(\frac{\pi }{4},0,\frac{3\pi }{4},0\right)$ (see text for the definition of the angles $\theta_{k}$ and $\phi_{K}$, k=1,2).

**Figure 2 entropy-25-00097-f002:**
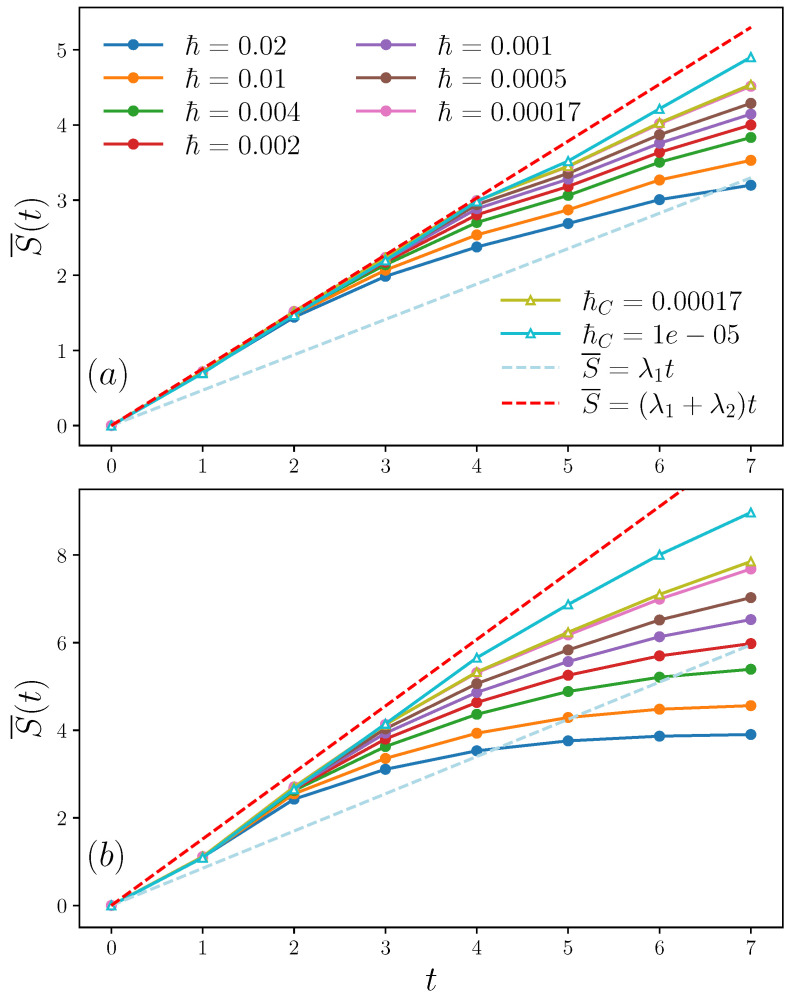
Quantum (circles with solid line) and classical (triangles with solid line) averaged linear entropy for different *ℏ* and $\hbar_{c}$ in the kicked coupled tops model defined in Equation (Equation 29), for (**a**): a=5, c=3 and (**b**): a=5, c=5. The dashed lines indicate the functions $\bar{S}=\left(\lambda_{1}+\lambda_{2}\right)t$ (red) and $\bar{S}=\lambda_{1}t$ (light blue). The initial values of S(t=0) are subtracted.

**Figure 3 entropy-25-00097-f003:**
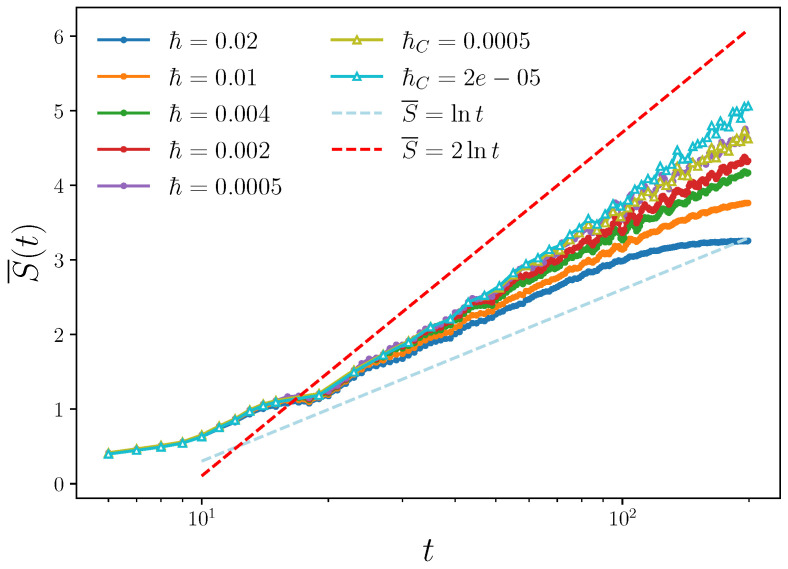
Same as in Figure 2, but for weaker coupling strength c=0.5, for which motion is quasi-integrable. The lines $\bar{S}\left(t\right)\propto \log t$ and $\bar{S}\left(t\right)\propto \log t^{2}$ are also drawn.

## Data Availability

The data are contained within the article.

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
