# Peer review of "Entanglement Dynamics and Classical Complexity"

_entropy, 2023, doi:10.3390/e25010097_

Round 1
Reviewer 1 Report
This addresses the problem of entanglement in quantum chaotic systems and concludes that indeed the Kolmogorov -Sinai entropy governs the growth of entanglement. The paper is well-written and supported by numerical evidence. It includes references that are crucial, but does miss out relevant references while including more than necessary perhaps in slightly unrelated topic of OTOC. I find the work interesting and goes towards clarifying a somewhat tricky issue. I recommend its publication, but would like the authors to address the following:
1. The analysis done is for the linear entropy, however many of the early studies and arguments are for the von Neumann entropy. Can the authors comment on the various other measures and especially this one.
2. The analysis in Ref. 23 is for linear harmonic /inverted oscillators. However, the analysis in the present paper is also in the same regime wherein the linearized dynamics is alone used, and Gaussians remain Gaussians. This being so, it is unclear what the novelty is. Also, it begs the question of whether the identification of the rates with the KS entropy is justified as this is an infinite time property, depending on the Lyapunov exponents. For linear systems, there is nothing else.
Author Response
We thank the reviewer for the report, helpful comments and her/his recommendation for publication.
Comment:
"This addresses the problem of entanglement in quantum chaotic systems and concludes that indeed the Kolmogorov -Sinai entropy governs the growth of entanglement. The paper is well-written and supported by numerical evidence. It includes references that are crucial, but does miss out relevant references while including more than necessary perhaps in slightly unrelated topic of OTOC. I find the work interesting and goes towards clarifying a somewhat tricky issue. I recommend its publication, but would like the authors to address the following"
Our reply:
We thank the reviewer for the encouraging comments. To the best of our knowledge we included all relevant publications but it might be possible that we have missed relevant references. In that case we apologize for that, however, since the referee does not provide specific suggestions, we do not know which references we need to add.
Comment:
"1. The analysis done is for the linear entropy, however many of the early studies and arguments are for the von Neumann entropy. Can the authors comment on the various other measures and especially this one."
Our reply:
We have added a paragraph at the beginning of Sec. 2 motivating our choice of the linear entropy, one being its convenience for numerical and analytical investigations, another one that we obtain at the same time very useful insights on the entanglement content of a quantum system.
Comment:
"2. The analysis in Ref. 23 is for linear harmonic /inverted oscillators. However, the analysis in the present paper is also in the same regime wherein the linearized dynamics is alone used, and Gaussians remain Gaussians. This being so, it is unclear what the novelty is. Also, it begs the question of whether the identification of the rates with the KS entropy is justified as this is an infinite time property, depending on the Lyapunov exponents. For linear systems, there is nothing else.".
Our reply:
Our theoretical analysis is valid up to the Ehrenfest time, up to which the quantum wave packet remains narrow and follows a beam of classical orbits. During this time interval the quantum wave-packet motion can be exponentially unstable as is the underlying classical trajectory, so chaos is included in our analysis. Moreover, as written in our conclusions, the Ehrenfest time diverges when the effective Planck constant goes to zero, and therefore it becomes appropriate to speak about entanglement growth rate with the classical KS entropy for longer and longer times.
Reviewer 2 Report
This is a nice work and absolutely appropriate to celebrate Prof. Casati 80th birthday.
I have no comments on the science nor on the presentations, and I recommend this paper for publication.
I would just like to point out 3 possible typos:
"constants of the motion" -> remove "the"
"The volume occupied by tori" -> "by the tori"
"phase spece distribution" -> "phase space distribution"
Author Response
We thank the reviewer for the report, and her/his recommendation for publication.
We have fixed the typos pointed out by the reviewer.